Fluoxetine-induced alteration of murine gut microbial community structure: evidence for a microbial endocrinology-based mechanism of action responsible for fluoxetine-induced side effects

Lyte Mark mlyte@iastate.edu 1
Daniels Karrie M. 1
Schmitz-Esser Stephan 2
1 Department of Veterinary Microbiology and Preventive Medicine, Iowa State University , Ames , IA , United States of America
2 Department of Animal Science, Iowa State University , Ames , IA , United States of America
Maggi Laura
Electronic publication date: 2019 Jan 9
Publication date: 2019
Volume: 7
Electronic Location ID: e6199
Received 2018 Sep 13; Accepted 2018 Dec 2
Copyright: ©2019 Lyte et al.
Copyright year: 2019
Copyright holder: Lyte et al.
License: This is an open access article distributed under the terms of the Creative Commons Attribution License, which permits unrestricted use, distribution, reproduction and adaptation in any medium and for any purpose provided that it is properly attributed. For attribution, the original author(s), title, publication source (PeerJ) and either DOI or URL of the article must be cited.
License URL: https://creativecommons.org/licenses/by/4.0/

Keywords: Fluoxetine, SSRIs, Depression, Lactobacillus, Microbial communities, Microbiota-gut-brain axis

Funding: United States Department of Defense, Office of Naval Research N000141512706 Iowa State University Foundation This work was supported by the United States Department of Defense, Office of Naval Research (No. N000141512706; to Mark Lyte) and the Iowa State University Foundation W. Eugene Lloyd Chair in Toxicology (to Mark Lyte). The funders had no role in study design, data collection and analysis, decision to publish, or preparation of the manuscript.

==============================
Background

Depression and major depressive disorder affect 25% of the population. First line treatment utilizing selective serotonin reuptake inhibitors (SSRIs) have met with limited success due to well-recognized negative side effects which include weight gain or loss. This inability to control unwanted side effects often result in patients stopping their antidepressant medications. The mechanisms underlying the failure of SSRIs are incompletely understood.

Methods

Male CF-1 mice (5 weeks of age, N = 10 per group) were per orally administered fluoxetine (20 mg per kg body weight) or diluent daily for 29 days. During this time fecal specimens were collected at three defined time points (0, 15 and 29 days). At the conclusion of the 29-day dosing regimen, animals were subjected to two behavioral assessments. For bacterial identification of the microbiota, 16S rRNA gene sequencing was performed on 60 fecal specimens (three specimens per mouse time course, N = 20 mice) using Illumina MiSeq. Analysis of community sequence data was done using mothur and LEfSe bioinformatic software packages.

Results

Daily per oral administration of fluoxetine for 29 days to male mice resulted in a significant, time dependent, alteration in microbial communities accompanying changes in body weight. The calculated species richness and diversity indicators of the murine fecal microbial communities were inconsistent and not significantly different between the groups. Among the phylotypes decreased in abundance due to fluoxetine administration were Lactobacillus johnsonii and Bacteroidales S24-7 which belong to phyla associated with regulation of body mass. The observed changes in body weight due to fluoxetine administration mimicked the dramatic shifts in weight gain/loss that has been observed in humans. Further, at the conclusion of the 29-day dosing regimen fluoxetine-dosed animals evidenced a mild anxiogenic-like behavior.

Discussion

We report that the most widely used antidepressant, fluoxetine, which is an SSRI-type drug, results in the selective depletion of gut microbiota, specifically the Lactobacilli which are involved in the regulation of body weight. Concomitantly, fluoxetine administration increases the abundance of phylotypes related to dysbiosis. Since Lactobacilli have been previously shown to possess a known biogenic amine transporter that regulates the uptake of fluoxetine, it is proposed that a microbial endocrinology-based mechanistic pathway is responsible for the ability of SSRIs to selectively negatively impact beneficial microbiota. The results of this study therefore suggest that the negative clinical side effects due to fluoxetine administration may be due to alterations in gut microbiota. Further, the data also suggests that supplementation of bacterial genera directly affected by fluoxetine administration may prove useful in ameliorating some of the well-known side effects of chronic fluoxetine administration such as weight alterations.

Introduction

The failure of drug therapy to adequately treat depression in large segments of the population and the numerous side effects that accompany its chronic administration such as anxiogenic effects and changes in body weight has been well recognized for years in both human (Anderson, 1998; Cascade et al., 2010) as well as animal populations (Aggarwal et al., 2016; Bagdy et al., 2001). Antidepressants, notably the selective serotonin reuptake inhibitors , otherwise referred to as SSRIs, are considered the first line of treatment for anxiety and major depressive disorder (Williams et al., 2017). SSRIs, as with the great majority of drugs, are taken per orally. Although the pharmacokinetic analysis of per oral dosing of SSRIs has demonstrated that the majority is absorbed into the host with peak plasma concentration occurring within 4–6 h following, a significant percentage of the administered per oral dose can in fact be recovered in the feces as has been shown in human volunteers (Lemberger et al., 1985). Thus, in addition to the host, the microbiota also represents a biological entity that interacts with any administered drug such as SSRIs. However, a crucial difference is that far less is known concerning the consequences of antipsychotic drug-microbiota interactions than are known for the host. Little, if any, research has been done on the mechanisms governing the effects of antidepressants on the microbiota and whether the failure and side effects of antidepressant therapy may be due to altered drug-induced microbiota. That it is entirely plausible that the microbiota composition, and in turn function, can be influenced by oral antipsychotic drugs can be seen in published evidence that has shown that among the large diversity of non-antibiotic drugs administered within the general population, antipsychotic drugs had potent effects on fecal microbial diversity (Maier et al., 2018). Additionally, a recent study analyzed the effect of fluoxetine on gut microbiota in rats (Cussotto et al., 2018), but so far, the effect of fluoxetine on gastrointestinal tract microbiota composition remains largely unknown.

In proposing a direct microbial effect of SSRIs, we have employed a microbial endocrinology-based theoretical framework as it has as its basis the concept of shared neurochemistry between microbes and host enabling bi-directional communication (Lyte, 2014). This approach has led to the recent discovery that selective bacterial genera within the gut microbiota, such as the Lactobacilli, possess the same two biogenic amine transports which are found in mammalian neuronal cells and are intimately involved in the therapeutic action of SSRIs (Lyte & Brown, 2018). The plasma membrane monoamine transporter (PMAT)- and the organic cation transporter (OCT)-like were shown to be present in Lactobacillus salivarius, but not L. rhamnosus, and display similar activity in regard to reuptake inhibitors such as the SSRI fluoxetine as is observed in mammalian neuronal cell cultures (Lyte & Brown, 2018).

As such, we hypothesized that per oral, chronic administration of fluoxetine could influence the diversity of the gut microbiota through direct microbial endocrinology-based interactions. This interaction would, by definition, be specific to only those bacterial genera which exhibit cell-based mechanisms, such as biogenic amine transporters, with which to interact with fluoxetine. This differs from previous reports which extend back decades that have reported that a number of psychoactive drugs can negatively influence bacterial viability and even suggested as an adjunct to therapy of a number of infectious diseases as the addition of an SSRI could lower the minimum inhibitory concentration of an antimicrobial with certain multi-drug resistant pathogens (Munoz-Bellido, Munoz-Criado & Garcia-Rodriguez, 1996; Munoz-Bellido, Munoz-Criado & Garcia-Rodriguez, 2000). The present study was designed to examine if a genus-specific effect of fluoxetine administration could be shown. If so, it then would become plausible to suggest that the ability of fluoxetine to influence host behavior as well as engender unwanted side effects could be due to direct microbial endocrinology-based effects within the host gut microbiota.

Materials and Methods

Animals

Twenty male CF-1 mice at 5 weeks of age (Charles River Laboratories, Wilmington, MA, USA) were randomized upon receipt and then housed at a density of 2 per cage (McCafferty et al., 2013). All experimental procedures were approved by the Iowa State University Institutional Animal Care and Use Committee, protocol #1-17-8420-M. Mice were weighed every other day in order to ensure that the correct mg per kg dosage was being administered. As shown (Fig. 1), body weight changed rapidly in response to fluoxetine administration.

Figure 1 (A) Weight changes over time in control and fluoxetine-dosed mice (N = 10 per group) and (B) Percent changes in bodyweight over time in control and fluoxetine-dosed animals (N = 10 per group).

Statistical analyses were conducted and p values are shown in the figure (two-tailed t tests).

Drug administration

Fluoxetine (TCI Chemicals, product #F0750, Portland, OR, USA) was initially dissolved in PBS at a concentration of 5 mg per ml. The resulting stock fluoxetine solution, as well as the diluent PBS, was then frozen in separate aliquots at −80 °C to provide for consistent solutions that were used for daily dosing. On the day of dosing, individual aliquots were thawed and administered to mice by gavage (flexible plastic feeding needles #9921, Cadence Science, Cranston, RI, USA) to achieve a concentration of 20 mg per kg body weight with equal volumes of diluent administered to control animals. This dosage was selected according to the landmark study of Dulawa et al. (2004) which evaluated the dose response relationship of chronic per oral fluoxetine administration in four strains of mice over an approximately 24 day period in various models of anxiety and depression and determined that 18 mg per kg body weight was found to be active in all behavioral paradigms.

Behavioral procedures

At the conclusion of the 29-day dosing regimen, two behavioral assessments, elevated-plus maze (EPM) and open field (OF), were conducted. These tests have been well-characterized to measure anxiety-like behavior in mice (Prut & Belzung, 2003; Walf & Frye, 2007). All movement and behavioral activity on the EPM and OF was digitally recorded using an HD 1,080 p webcam (Logitech, Newark, CA) coupled to a Windows-based computer running the Any-maze behavioral tracking software (Stoelting Co., Wood Dale, IL, USA). Statistical analyses of the behavioral data was performed using GraphPad Prism statistical software package (version 7.05, GraphPad Software, La Jolla, CA, USA).

DNA isolation, MiSeq sequencing and sequence analyses

Immediately following the last behavioral measure, mice were sacrificed via cardiac puncture. Fecal pellets were removed from the large intestine and from the behavioral testing device immediately prior to sacrifice and stored at −80 °C. Genomic DNA isolation was obtained using the PowerSoil DNA Isolation Kit (MoBio, Carlsbad, CA, USA).

In total, 60 fecal samples, representing 20 samples for each time point (day 0, 15, and 29) with 10 samples for the control and treatment groups, respectively, were used for 16S rRNA gene amplicon sequencing using Illumina MiSeq with 151 bp paired-end sequencing technology. 16S rRNA gene PCR and library preparation and sequencing was completed at the Environmental Sample Preparation and Sequencing Facility at Argonne National Laboratory. For Illumina sequencing, Genomic DNA was amplified using the Earth Microbiome Project barcoded primer set, adapted for the Illumina MiSeq by adding nine extra bases in the adapter region of the forward amplification primer that support paired-end sequencing. The V4 region of the 16S rRNA gene (515F-806R) was amplified with region-specific primers that included the Illumina flowcell adapter sequences. The reverse amplification primer contained a twelve base barcode sequence. Each 25 µl PCR reaction contained 12 µl of MoBio PCR Water (Certified DNA-Free), 10 µl of 5 Prime HotMasterMix (1×), 1 µl of Forward Primer (5 µM concentration, 200 pM final), 1 µl Golay Barcode Tagged Reverse Primer (5 µM concentration, 200 pM final), and 1  µl of template DNA. The conditions for PCR were as follows: 94 °C for 3 min, with 35 cycles at 94 °C for 45 s, 50 °C for 60 s, and 72 °C for 90 s; with a final extension of 10 min at 72 °C. The PCR amplicons were quantified using PicoGreen (Invitrogen) and a plate reader. Once quantified, different volumes of each of the products were pooled into a single tube so that each amplicon is represented equally. This pool was then cleaned up using UltraClean® PCR Clean-Up Kit (MoBio), and then quantified using the Qubit (Invitrogen). After quantification, the molarity of the pool was determined and diluted down to 2 nM, denatured, and then diluted to a final concentration of 6.75 pM with a 10% PhiX spike for sequencing on the Illumina MiSeq. Sequence analysis was performed using mothur version 1.39.3 following the mothur MiSeq SOP available at the mothur website (Kozich et al., 2013). Briefly, contigs were joined with “make.contigs”, reads longer than 250 bp, harboring any ambiguous bases or with more than 8 consecutive homopolymers were excluded using “screen.seqs”. Chimeric sequences were removed with “chimera.uchime” and the reads were clustered into operational taxonomic units (OTUs) using a 97% similarity threshold and taxonomy was assigned to OTUs using the SILVA NR 128 reference database (Quast et al., 2013). To identify biomarkers that differ in abundance between groups were done with the LEfSe (Segata et al., 2011) implementation in mothur; for this, p-values <0.05 were considered significant. As a first step, LEfSe performs a Kruskall-Wallis test to analyze all features whether the values in different classes are differentially distributed. In a second step, a pairwise Wilcoxon test is performed with the retained features. In the last step, a linear discriminant analysis model is built from the retained features to determine the effect sizes for each feature. For determination of differences between groups on the community level, analysis of molecular variance (AMOVA) and analysis of similarity (ANOSIM) implemented in mothur was used. Heatmaps were generated with JColorGrid (Joachimiak, Weisman & May, 2006). For better taxonomic classification, the OTUs were searched against the 16S rRNA gene sequences of the Mouse Intestinal Bacterial Collection isolates (miBC; Lagkouvardos et al., 2016) using BlastN.

Results

As can be seen in Fig. 1, animals administered 20 mg/kg body weight of fluoxetine over a 29-day dosing period evidence changes in body weight while those administered diluent did not. This dosage, as previously discussed in the ‘Materials and Methods’ section, was chosen based upon a landmark study examining the dose response relationship of chronic per oral administration in three strains of mice and its ability to influence models of anxiety and depression (Dulawa et al., 2004). In Fig. 1A, the ability of fluoxetine to differentially increase or decrease weight in select animals is evident while such dramatic shifts were not observed in any of the control animals. As shown in Fig. 1B, fluoxetine administration resulted in a statistically significant (p = 0.018, two-tailed t-test) change in weights gain/loss during the first 15 days. This result has also been observed in humans and animal models as administration of fluoxetine can lead to dramatic changes in weight (Anderson, 1998; Cascade et al., 2010). Firmicutes were more abundant than Bacteroidetes in the fluoxetine treated mice (Fig. 2). As such, there was no consistent change in one direction of weight gain or loss that could be correlated with the Firmicutes/Bacteroidetes ratios which has also been noted in humans (Sze & Schloss, 2016). This observation of fluoxetine-induced changes in mice is also in agreement with observations in the human population that it is a change in body weight, which could be either a gain or loss, which is experienced in individuals administered given fluoxetine. Currently, there are no predictive measures for the a priori prediction of the direction of the weight change.

Figure 2 Relative abundance of bacterial phyla.

The figure shows mean relative abundance values for Firmicutes, Bacteroidetes, Tenericutes, and Proteobacteria in the control group and fluoxetine-treated mice. Error bars represent standard error of the mean.

As shown in Fig. 3, behavioral testing of mice tested at the end of the 29-day dosing period revealed the development of mild anxiogenic-like behavior. For the EPM an increase in entries into the closed arm (Fig. 3A; p = 0.047) as well as decreased time in the center platform of the EPM (Fig. 3B; p = 0.030) were observed. For the OF an increase in total number of rearings in all zones (Fig. 3C; p = 0.008), time rearing in the periphery zone (Fig. 3D; p = 0.009), distance traveled in the periphery (Fig. 3E; p = 0.042) were observed while decreased time in the center zone (Fig. 3F; p = 0.035) was seen. These fluoxetine anxiogenic-like induced effects are also consistent with what has been observed in human and animal models (Aggarwal et al., 2016; Anderson, 1998; Cascade et al., 2010).

Figure 3 Behavioral assessment of control (N = 10 per group) and fluoxetine-dosed (N = 10 per group) mice at conclusion of 29-day dosing regimen.

(A) Entries into closed arm of EPM; (B) time in center platform of the EPM; (C) total rearings in all zones of the OF; (D) time rearing in the periphery zone of the OF; (E) distance traveled in the periphery zone of the OF; and (F) time in the center zone of the OF. Statistical analyses were conducted and p values are shown in the figure (two-tailed t tests).

In total, 1.42 million reads were obtained after merging the forward and reverse reads. After quality control, 1.134 million high quality reads remained which were clustered into 1,612 OTUs with at least 10 reads. On phylum level, the microbiota of the mice in this experiment was dominated by Firmicutes (51.4%), Bacteroidetes (44.8%), Tenericutes (1.2%), and Deferibacteres (1.2%) (Fig. 2). All other phyla showed less than 1% relative abundance among all samples.

The calculated species richness and diversity indicators were inconsistent and not significantly different between the groups: Some animals showed an increase in species richness and diversity during the trial, and others stayed similar or decreased (Table S1). No significant differences (p > 0.57) were found in the richness and diversity estimators between the control and the fluoxetine treated group. Most of the OTUs had highest similarity to phylotypes described as members of the murine gastrointestinal tract (Fig. 4, Tables S2, S3). Our results revealed clear differences in abundance of OTUs between the two groups. Among all OTUs, 121 OTUs were significantly different between control and the fluoxetine group; 21 of these OTUs were among the 100 most abundant OTUs (Table S4). Some Bacteroidales S24-7 group, some Lachnospiraceae OTUs and the Lactobacillus OTU17 (the latter showed a 7.4-fold decrease under fluoxetine treatment) were significantly more abundant in the control group, whereas the Alistipes OTU24, the Lachnoclostridium and Anaerotruncus OTUs were more abundant in the fluoxetine-treated mice.

Figure 4 Relative abundance of the 30 most abundant OTUs.

The heatmap shows median relative abundance values for OTUs in the control group and fluoxetine-treated mice. OTUs which were statistically significantly different between the two groups based on LEfSe (Segata et al., 2011) are highlighted in bold and by asterisks (see Table S4 for details).

On the genus level, Bacteroidales S24-7 group, unclassified Lachnospiraceae, and Lachnospiraceae_NK4A136 group were most abundant (Fig. 5). Similar to our findings on OTU level, a genus belonging to the Bacteroidales S24-7 group and genera affiliating to the Ruminococcaceae_UCG-014 group were significantly more abundant in the control group (Table S5). Genera affiliating to the Lachnospiraceae_UCG-001 and UCG-006 groups, to Anaerotruncus, Lachnoclostridium and to uncultured Lachnospiraceae showed significantly higher abundance in the fluoxetine treated mice. In addition, also Anaerotruncus, Ruminiclostridium_5, unclassified Coriobacteriaceae, and Lachnoclostridium were significantly more abundant in the fluoxetine treated mice. Comparisons on whole community level using AMOVA and ANOSIM revealed significant differences between the microbial communities of the control group and the fluoxetine treated mice (p = 0.02 and p = 0.05, R = 0.28, respectively).

Figure 5 Relative abundance of the 20 most abundant genera.

The heatmap shows median relative abundance values for genera in the control group and fluoxetine-treated mice. Genera which were statistically significantly different between the two groups based on LEfSe (Segata et al., 2011) are highlighted in bold and by asterisks (See Table S5 for details).

Discussion

Overall, our data revealed significant differences between the microbial communities of mice from the control group compared to the fluoxetine treated mice indicative of a shift of microbial communities towards dysbiosis induced by the fluoxetine treatment. As discussed in prior sections, we chose to administer a chronic dose of fluoxetine (20 mg per kg of body weight) that had been previously shown in a landmark study examining the dose response relationship of chronic fluoxetine administration in four strains of mice to be the only one active in all employed models of anxiety and depression (Dulawa et al., 2004). Specifically, a dose response range from 0 to 25 mg per kg body weight was employed with the finding that 18 mg per kg was the only dosage found effective in all behavioral paradigms (Dulawa et al., 2004). As can be seen in Fig. 1, statistically significant changes in bodyweight occurred during the first 15 days of administration that closely mimic what is observed in humans and other animal models. Such dramatic shifts in weight gain/loss are often cited as a primary reason for patient non-compliance in continuing drug therapy (Anderson, 1998; Cascade et al., 2010). The ability of this chronic dose of fluoxetine to influence standard measures of anxiety-like behavior in mice was also observed (Fig. 3) as has been noted in other studies which have employed chronic administration of similar levels of per oral administered fluoxetine (Dulawa et al., 2004; Gosselin et al., 2017).

The bioinformatic analyses examining the ability of fluoxetine to influence the composition of the microbial communities demonstrated novel effects not previously reported in the literature. Most strikingly, the analyses revealed that some OTUs were significantly higher abundant in the control group, such as several Bacteroidales S24-7 group OTUs and one Lactobacillus OTU. Recently, an inhibition of growth of Lactobacillus by fluoxetine has been described (Cussotto et al., 2018) and Lactobacillus treatment reduced depressive-like behavior (McVey Neufeld, Kay & Bienenstock, 2018). It should be noted that not all Lactobacillus OTUs in our study decreased due to the fluoxetine treatment. This could suggest that the fluoxetine treatment might affect different Lactobacillus strains or species differently. This differential effect of fluoxetine on lactobacilli is especially intriguing in light of the reports that demonstrate that specific species and strains, such as L. reuteri (Marin et al., 2017) and L. rhamnosus JB-1 (McVey Neufeld, Kay & Bienenstock, 2018) can influence behavior, while other lactobacilli cannot (Bravo et al., 2011; Vancanneyt et al., 2006). A number of OTUs and genera were significantly more abundant in the treatment group included Alistipes, various Lachnospiraceae OTUs, Lachnoclostridium and Anaerotruncus. Alistipes belongs to the Bacteroidetes and is often found in murine gastrointestinal tract microbiota samples (Lagkouvardos et al., 2016). Alistipes have recently been found to be increased in abundance in patients with gastrointestinal complications after thoracic aortic dissection surgery (Zheng et al., 2017), and are reported to be associated with human colorectal carcinoma (Shi et al., 2017). Alistipes have been shown to induce colitis and tumors in mice (Moschen et al., 2016) and were increased in mice exposed to ethanol (Peterson et al., 2017). Alistipes OTU24 might thus be an indicator of gastrointestinal dysbiosis. However, other Alistipes OTUs found in this study (e.g., OTUs 23, 25, 47) did not reveal significant differences in abundance between the two experimental groups. A recent study has shown that Alistipes was reduced in abundance in mice in response to chemically or pathogen induced colitis and this study indicated that Alistipes is particularly sensitive to inflammation and possesses butyrate production capacity (Borton et al., 2017). This suggests that different phylotypes/OTUs within a genus or family can possess different metabolic properties as suggested earlier (Berry et al., 2012). Anaerotruncus is a genus in the Firmicutes, the function of its members remains largely unknown. Bacteria affiliating to the genus Anaerotruncus have been isolated from fecal samples from patients with obesity or malnutrition (Pham et al., 2017; Togo et al., 2016) and have been associated with bacteremia in humans (Lau et al., 2006). Furthermore, Anaerotruncus was increased in rats exposed to prenatal stress (Golubeva et al., 2015). These data indicate that Anaerotruncus may be an indicator of gastrointestinal dysbiosis. Several OTUs affiliating to the Lachnospiraceae family (OTUs 32, 38, 86, 93) were significantly more abundant in the fluoxetine treated mice. Our knowledge about the metabolic properties of many members of the Lachnospiraceae is still limited. A recent study has revealed members of unclassified Lachnospiraceae as important drivers of gastrointestinal dysbiosis in mice (Moschen et al., 2016).

Three OTUs affiliating the genus Roseburia (OTUs 45, 69, 74) showed significant differences between the two conditions. Two of these OTUs (45 and 74) were more abundant in the fluoxetine treated mice, whereas OTU69 was more abundant in the control group. In general, Roseburia is considered a beneficial commensal bacterium producing butyrate and having a positive impact on the host immune system and the abundance of Roseburia has been reported to decrease under various disease conditions (Patterson et al., 2017; Tamanai-Shacoori et al., 2017). However, recently, a higher abundance of Roseburia was linked to gut dysbiois in mice and to cerebral hypometabolism (Sanguinetti et al., 2018). The higher abundance of Roseburia OTUs 45 and 74 in the fluoxetine treated mice provides preliminary evidence that drug-induced changes in microbial communities may be a contributing factor in the development of altered physiology.

OTU46 is assigned to the genus Lachnoclostridium comprising members of the Clostridum cluster XIV including the former Clostridium scindens (Yutin & Galperin, 2013) and shows 99% 16S rRNA gene similarity with C. scindens ATCC35704. Several C. scindens and related Clostridum species are able to convert primary bile acids to toxic secondary bile acids such as deoxycholic acid and lithocholic acid which have been linked to diseases of the gastrointestinal tract such as liver and colorectal cancer (Ridlon et al., 2016). We thus speculate that the increase of OTU46 in the fluoxetine treated mice is a sign of dysbiosis, possibly by the production of toxic secondary bile acids.

The most abundant OTUs and genus found in this study belong to the Bacteroidales S24-7 group, including the recently described Muribaculum intestinale (Lagkouvardos et al., 2016). Members of the Bacteroidales group S24-7 are highly abundant particularly in murine and human gastrointestinal tract samples (Borton et al., 2017; Lagkouvardos et al., 2016; Ormerod et al., 2016). Both on OTU and genus level, Bacteroidales S24- 7 showed higher abundance in the control group samples. Recent evidence shows that members of the Bacteroidales S24-7 group are able to produce propionate and have mucin degrading capacity (Borton et al., 2017; Ormerod et al., 2016) suggesting a beneficial role for gastrointestinal tract health. In line with this, a reduction of the Bacteroidales S24-7 group has been described recently in colitic and obese mice (Huazano-Garcia, Shin & Lopez, 2017; Osaka et al., 2017). Furthermore, Bacteroidales S24-7 are reduced in mice prone to develop dementia, suggesting a possible link of Bacteroidales S24-7 with normal brain function (Sanguinetti et al., 2018).

The study by Cussotto et al. (Cussotto et al., 2018) showed that Prevotella and Scuccinivibrio are decreased by fluoxetine. In our study, we did not identify Prevotella or Succinivibrio among the 100 most abundant OTUs, which may be explained by different experimental settings between the two studies such as different fluoxetine concentrations and different animals (rat and mice).

The governing hypothesis in the present manuscript is based on the shared neurochemistry between microbes and host, what has become known as microbial endocrinology (Lyte, 2014). As shown, we have obtained the first data demonstrating that microbiota belonging to a specific genera, such as Lactobacillus, and which have previously been shown to possess the biogenic amine transporters PMAT and OCT (Lyte & Brown, 2018), can be influenced by administration of an SSRI. It should however, be noted that the genes encoding the Lactobacillus PMAT and OCT transporters have not been identified yet. The ability of fluoxetine to affect the abundance of the other genera discussed above raises the possibility that they may also possess biogenic amine transporters similar to that of Lactobacilli. Our lab is currently examining this possibility, but research is hampered by the fact that representative members of fluoxetine-affected genera in this study are not available in microbial culture collections. The physiological role of PMAT- and OCT-like transporter activity in Lactobacilli is not yet understood.

It is critical to separate the results, and implications, of the present study from that of others in which direct antimicrobial effects of SSRIs were observed (Munoz-Bellido, Munoz-Criado & Garcia-Rodriguez, 1996; Munoz-Bellido, Munoz-Criado & Garcia-Rodriguez, 2000). In those studies, antimicrobial effects were observed in an in vitro assay system where there was no possibility for concentrations of fluoxetine that exceed what can be achieved in vivo to be eliminated. Further, bacterial genera that were observed to be susceptible to mM concentrations of fluoxetine, such as Escherichia, were not observed to be decreased in the present study. As such, the working hypothesis is that fluoxetine is exerting its effects through a PMAT- and OCT-like transporter in affected genera for which the precise physiological role of this transporter has yet to be defined. The present data conclusively demonstrates a genus-specific effect of fluoxetine that cannot be understood in the context of solely antimicrobial activity for which concentrations needed to achieve an either bacteriostatic or bactericidal effect need to be much higher than those that can be achieved in the gut. Given the recent publication which has shown that in humans the administration of antipsychotics disrupts fecal microbial diversity (Maier et al., 2018), it is of importance to understand the mechanisms by which such alterations occur and if they impact drug efficacy.

The role of the microbiota, and community structure, in determining behavior has come under increasing investigation (Foster et al., 2016). A bi-directional axis involving the microbiota, gut and brain has been proposed in which the microbiota influence behavior, although mechanistic studies are still severely lacking. The need for identification of mechanisms will be crucial if the concept of a microbiota-gut-brain axis is to progress especially as regards eventual clinical application. The present study has shown that a microbial endocrinology-based mechanism, namely on involving the PMAT- and OCT-like biogenic amine transporters, may be one of the mechanisms responsible. The observed differential changes in the abundance of specific microbiota genera that are well correlated (as has been discussed above) with gut dysbiosis, suggests that the well-recognized negative side effects of fluoxetine may be due to changes in gut microbiota that are mediated via a bacterial biogenic amine transporter. The results of this study further suggest that restoration of physiologically beneficial microbial diversity may have therapeutic potential to ameliorate some of the negative side effects of fluoxetine thereby increasing drug efficacy and patient compliance.

Conclusions

The present study has demonstrated that mice per orally administered fluoxetine exhibit specific alterations in microbial community structure concomitant to alterations in body mass as well as the development of mild anxiogenic-like behavior. As this study was undertaken to examine the possible role of the microbiota as a contributing factor in the development of the unwanted side effects with the use of fluoxetine in the treatment of anxiety-related illness in humans, this report has provided first known initial evidence for the involvement of a specific genus, Lactobacillus, that is also known to be involved in regulation of body mass. Further, as prior work has shown that Lactobacilli possess a biogenic amine transporter capable of incorporating fluoxetine (Lyte & Brown, 2018), it is proposed that a microbial endocrinology-based specific mechanistic pathway is, in part, responsible for the fluoxetine-induced weight alterations. By demonstrating the ability of fluoxetine to affect the microbial community structure in a defined manner, the possibility that restoration of normal structure through the use of probiotics may provide a means by which the negative clinical side effects of fluoxetine leading to patient non-compliance in the treatment of depression may be ameliorated.

Supplemental Information

Table S1 Species richness and diversity estimators

Click here for additional data file.

Table S2 The 50 most abundant OTUs across all samples

Click here for additional data file.

Table S3 BlastN comparison of OTUs with Mouse Intestinal Bacterial Collection (miBC) 16S rRNA gene sequences

Click here for additional data file.

Table S4 Significantly different OTUs between the two groups determined by LEfSe implemented in mothur

Significantly different OTUs are shown for the 100 most abundant OTUs.

Click here for additional data file.

Table S5 Significantly different genera (p < 0.05) between the two groups determined by LEfSe implemented in mothur

Significantly different genera are shown for the 50 most abundant genera.

Click here for additional data file.

The Authors would like to thank Dr. Peter Clark (Iowa State University) for helpful discussions regarding the use of behavioral testing in relationship to antidepressant administration.

Additional Information and Declarations

Competing Interests

Author Contributions

Animal Ethics

Data Availability

The authors declare there are no competing interests.

Mark Lyte conceived and designed the experiments, analyzed the data, contributed reagents/materials/analysis tools, authored or reviewed drafts of the paper, approved the final draft.

Karrie M. Daniels performed the experiments, analyzed the data, prepared figures and/or tables, authored or reviewed drafts of the paper, approved the final draft.

Stephan Schmitz-Esser performed the experiments, analyzed the data, contributed reagents/materials/analysis tools, prepared figures and/or tables, authored or reviewed drafts of the paper, approved the final draft.

The following information was supplied relating to ethical approvals (i.e., approving body and any reference numbers):

Institutional Animal Care and Use Committee of Iowa State University provided full approval for this animal experiments (1-17-8420-M).

The following information was supplied regarding data availability:

The sequencing data is available at the NCBI Sequence Read Archive SRA under accession number SRP145610.

It is also available at: https://www.ncbi.nlm.nih.gov/Traces/study/?acc=SRP145610.

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
