# Peer review of "Fluoxetine-induced alteration of murine gut microbial community structure: evidence for a microbial endocrinology-based mechanism of action responsible for fluoxetine-induced side effects"

_PeerJ, doi:10.7717/peerj.6199_

## Round 0.1 · original submission · Major Revisions

I suggest to re-organize results in clear and separate sections. Behaviour results should be included in the discussion or removed and the statistical approach used in the behavioural characterisation should be specified. In addition, please rephrase the sentence: "A numerical trend, although not statistically significant (p=0.08)" , the trend clearly indicate an absence of statistical significance.

Reviewer 1 ·

Basic reporting

A very interesting and relevant work. The sample size is large and looks very elaborate.

Experimental design

The experiments are well detailed.

Validity of the findings

Has a microbiota markers been used to compare groups as an endogenous marker?

Additional comments

Well detailed and consistent manuscript.

Reviewer 2 ·

Basic reporting

no comment

Experimental design

no comment

Validity of the findings

no comment

Annotated reviews are not available for download in order to protect the identity of reviewers who chose to remain anonymous.

Reviewer 3 ·

Basic reporting

Well written and appropriately structured manuscript.

Experimental design

Sufficient level of detail in methods and satisfactory experimental design. Concern surrounding reproducibility. Ten per test group is quite low when looking at such variable measures as behavior and microbiota.

Validity of the findings

Concern surrounding reproducibility. Ten per test group is quite low when looking at such variable measures as behavior and microbiota. It is remarkable that statistical significance was achieved. Although these dramatic effects many represent a big impact, published results should be the product of 3 independent experiments. So it is recommended that the study be repeated.

Additional comments

The manuscript is well written and clearly states aims and conclusions of study.
More ‘quantitative’ figures, outlining phyla abundances and OTUs should be included in the main figures, rather than as supplemental figures, rather than just the heat maps. Perhaps find another way than tables to represent these data.
Figure 1: Data should be presented as mean weight ± standard deviation. Fluoxotine vs. control should be directly compared at each time point and ANOVA applied to the data to determine whether any differences in overall weight are seen between groups. Add n= 10 per group and appropriate figure legend, use format similar to figure 2.

---

## Round 0.2 · accepted · Accept

The paper is now been improved following reviewers suggestions.
Since the the findings are rather limited, I suggest in the future to develop them in a more complete form to improve their potential impact.

# Reviewer 2 ·

Basic reporting

good

Experimental design

good

Validity of the findings

good

Additional comments

The authors have addressed all my concerns and the manuscript is now suitable for publication.

Reviewer 3 ·

Basic reporting

Well written. Satisfactorily responded to all reviewer comments.

Experimental design

Appropriate

Validity of the findings

As a pilot study ok, but findings should be reproduced.